



# 1   **Semi-automated roadside image data collection**

Neal Pilger[1], Aaron Berg[1], Pamela Joosse[2]
[1]Department of Geography, University of Guelph, Guelph, ON, N1G 2W1, Canada
[2]Agriculture and Agri-Food Canada, 174 Stone Road W., Guelph, ON NIG 4S9, Canada
*Correspondence to*: Neal Pilger (npilger@uoguelph.ca)
**Abstract.**
This article describes the development of a mobile roadside survey procedure for obtaining corroboration
data for the remote sensing of agricultural land use practices. The key objective was to produce a dataset of
geo-referenced roadside digital images that can be used to compare to in-field photos for measuring
agricultural land use and land cover associated with crop residue and cover cropping in the non-growing
season. It was concluded that a very high level of correspondence (>90% level of agreement) could be
attained using a mobile survey vehicle, as presented in this research, to detailed in-field ground verification
data. Classification correspondence was carried out against 114 field sites with a level of agreement at
93%. The few discrepancies were in the differentiation of residue levels between 30-60% and >60%, both
of which may be considered as achieving conservation practice standards.  The mobile roadside image
capture has advantages of relatively low cost and insensitivity to cloudy days, which often limits optical
remote sensing acquisitions during the study period of interest. We anticipate that this approach can be used
to reduce associated field costs for ground surveys, while expanding coverage areas and may be of interest
to industry, academic and government organizations for more routine surveys of agricultural soil cover
during periods of seasonal cloud cover.

## 22   Introduction

The identification and verification of in-field characteristics is an inherent component of the remote sensing
of land use and land cover (LULC), and change detection classifications for the assessment of post-harvest
tillage and cover crop practices (Hussain, *et al*., 2013). Traditionally, the generation of training and
evaluation data sets for remote sensing classification approaches rely on in-field physical measures that
include both nadir and oblique image capture, physical counts of residue to bare earth percentages over a
series of 3 - 5 geo-referenced sample plots to represent a satellite pixel (e.g. Pacheco and McNairn, 2010;
AAFC, 2011; Laamrani *et al*. 2017). Such methods, while effective in the categorization of crop residue
classes for research purposes, are costly, labour intensive, and limited logistically to  characterize a large





region, such as at the county level, in quantifying annual tillage and cover crop use and trends. These
methods are also inefficient for situations when generalized classes of landcover are sufficient for program
and policy decision making activities.
The common post-harvest activities used in agriculture land management in the southern Ontario study
region include conventional (CV), conservation (CS) and no tillage (NT), and potentially cover cropping
practices; defined as follows. Conventional tillage is a common post-harvest practice for many large-scale
agricultural operations. This tillage practice involves incorporating, or turning residual plant matter into the
soil following harvest, and with additional seedbed preparation prior to the following planting cycle.
Conventional tillage is effective at controlling weeds, however, the burial of residue, and the increased
disaggregation of the soil encourages runoff and erosion  (Moreira *et al*., 2016; Dam *et al*., 2005). This
practice also leads to an increase in carbon release to the atmosphere via accelerated organic soil matter
breakdown, which has been linked to climate change (Silva-Olaya *et* al., 2013). Aside from these issues is
the cost in time and fuel (if using mechanized tools – e.g. tractors) of repeated passes over the field.
Conservation tillage (CS) and No-tillage (NT) use implements designed to limit soil disturbance to reduce
surface disruption, and leave a protective organic layer (crop residue) between harvest and subsequent
plantings (Steiner, 2002). The difference between CS and NT residue classes lie in the amount of material
left between harvest and replanting on the surface, where CS residue is typically classified with residue
coverage between 30 and 60 percent, and NT categorization having in excess to 60 percent, as opposed to
residue cover significantly less than 30 percent for conventional (CV) practices. Both CS and NT practices
have been shown to influence soil microbial biomass (Mathew, *et al*., 2012; Govaerts *et al*., 2007;
Spedding *et al*., 2004) and hydraulic properties (Blanco-Canqui *et al*., 2017; Gozubuyuk *el al*., 2014) by
improving soil quality via an increase in soil organic matter (Daughtry and Hunt, 2008). Both methods
reduce soil disturbance, compared to conventional methods, and therefore assist in carbon sequestration
(Dolan et al., 2006; Halvorson *et al*., 2002; Angers *et al*., 1997). Maintaining large amounts of non-
photosynthetic plant material on the surface, no-till practices somewhat mimic a natural ecosystem scenario
(Jabro *et al*., 2016; Arshad *et al*., 1999).The added layer of plant material, however, can trap moisture and
create a fertile environment for both fungal and weed development (Govaerts *et al*, 2007).

Another practice  employed after crop harvest is to establish a living plant green cover.  In this study, green
cover in fields included fields planted to perennial crops (i.e. predominantly alfalfa or alfalfa/grass
mixtures), winter cereals (i.e. winter wheat predominantly) as well as cover crops (e.g. Oats, oilseed radish,
clover). The green cover secures the inverted soil against wind erosion and maintain moisture levels and

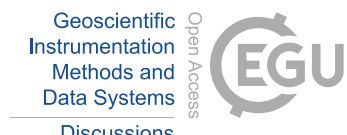

material for decomposition prior to spring planting. Green cover species with deep tap roots can be
important to breaking through compacted layers and may be considered a fourth tillage practice (Derpsch,
2003). If green cover can get established early enough that there is significant cover of the soil, it can
effectively dissipate direct rainfall striking the soil surface promoting diffuse infiltration and limiting
potential for surface runoff and erosion.
As of 2009, Agriculture and Agri-Food Canada (AAFC) implemented a crop inventory database in Canada
using satellite remote sensing data and ground-based verification (Fisettte *et* al., 2014). While orbital, or
high altitude aerial remote sensing is routinely employed for land-use classification, optical remote sensing
methods are limited by opaque atmospheres (Thoma, *et al*., 2004) as regularly present in both the  spring -
pre-planting; and in  fall, post-harvest. While Radar remote sensing has been utilized to circumnavigate
such issues, there are remaining issues in the collection of ground data to develop wider-scale
implementation of remote sensing classification approaches for non-growing season land management
practices. Planning field verification missions, being costly from a personnel and travel perspective, can be
minimized using single vehicle large scale surveys, which function equally well in clear or overcast
conditions. Roadside surveys, however, primarily focus on oblique horizontal/ landscape data capture, as
opposed to vertical nadir views afforded by most high altitude airborne and orbital platforms.

An issue prevalent in using oblique photography for landscape evaluation lies in the variable scale inherent
in such images as function of image tilt, focal length, sensor resolution, and camera height (Remondino and
Gerke, 2015). Subsequently, the background image pixels are representative of a larger geographic area
than their foreground counterparts, and effective quantification of surface variation is limited to relative,
rather than absolute measures (Stockdale, *et al*., 2015). For generally homogeneous landscape
categorization, however, especially if being used for the generation of ground-truth training sites against
nadir-view high altitude aircraft or satellite remote sensing classification, such methods are suitable for
rapid ground class assessment. The virtue in oblique imagery lies in its simplicity of interpretation and
understanding, being the way in which we are accustomed to viewing the world (Remondino and Gerke,

28  2015).


The objective of this study was to establish a baseline percentage of Ontario county level agriculture fields
employing different tillage practices through the development, and testing of a ground survey data, vehicle-
mounted, camera system.  The survey system was developed to compare to in-field photos for measuring
soil cover in order to determine the value of roadside acquisition for both routine ground truth data



collection for remote sensing analysis of soil cover, and the utility of using such data collection as a
surrogate to standard practices which are reliant on orbital remote sensing classification.
**Methodology**
**Site Location**
Fieldwork was conducted in  Elgin and Essex county in South-western Ontario, Canada (Figure 1).
Dominant crops in these areas include corn (*Zea mays*),  and soybean (*Glycine max*), and winter wheat
(*Triticum aestivum L.*) grown in rotation, interspersed occasionally with perennial forages. As discussed
above, various practices are followed after harvest including use of cover crops, conventional, CT and NT,
resulting in different soil surface cover conditions during the non-growing season.



**Figure 1. Map of roadside survey area locations.**

**Instrumentation**

The vehicle mounted road side imagery system (Figure 2) included a pair of Garmin VIRB XE cameras and a full-spectrum (near-infrared) modified GoPro HERO camera. The imaging systems were mounted perpendicular to the vehicles travel direction, extended above permanent mounting brackets to an elevation of 2.3m (7.5') on the right (curb-side) and up to 2.6m (8.5') on the left (driver-side) above ground level to compensate for the additional distance to drivers-side fields. The extension poles were further reinforced against vibration using foam insulated cable ties to roof rails, and support mounts on the vehicle. A large proportion of the driving route for each county was comprised of rough gravel, or packed dirt roads; hence a 4WD with a modified soft suspension was required to reduce vibration on the imaging platforms themselves. Further the 4WD vehicle with its higher ground clearance allowed for higher mounting of the side-looking cameras.

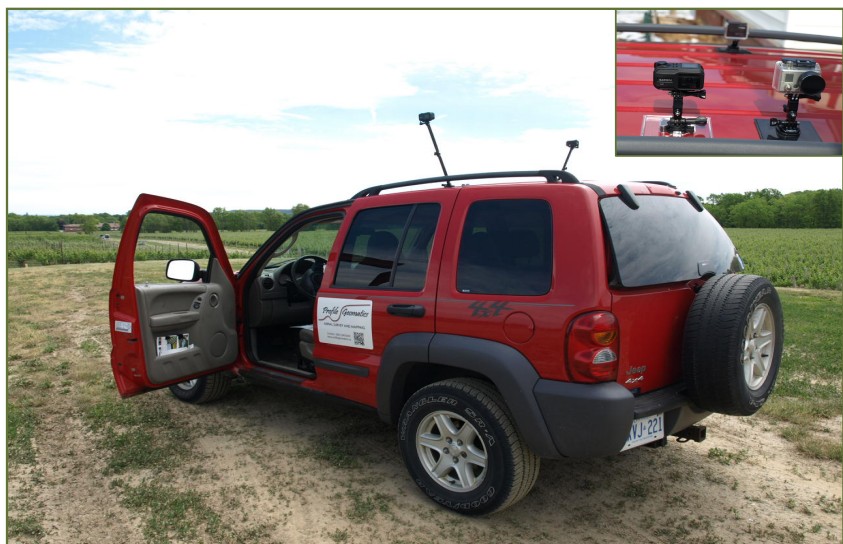

**Figure 2. Roadside survey vehicle camera system with roof-top camera mounts**

Data collection for Essex and Elgin County were carried out on May 18, and 20, 2016, respectively under clear sky conditions. Acquisition plans were conducted to coincide with Landsat 8 OLI overpass and the concurrent in-field ground-truth data collection for each respective area. Data collection was carried out in this manner to produce a value-added product that could be employed for ground-truth evaluation of future studies employing the Landsat orbital platform. Thus, permitting additional use for the collected imagery in



a secondary land-use study not covered in this pilot experiment.
**Table 1. Weather and sky conditions during the two roadside image acquisitions.**

| Location | Essex County | Elgin County |
|---|---|---|
| Date | 2016-05-18 | 2016-05-20 |
| Route Distance | 102 km | 248 km |
| Mean Temperature | 12 C | 14 C |
| Sky Conditions | mostly clear | mostly clear |
| Visibility | unlimited | unlimited |
| Pressure | 1021.92 hPa | 1023.07 hPa |

**Driving Parameters**
Route maps were designed for the two sampling locations to minimize overlap in image acquisition, and to
ensure adequate coverage of AAFC field plots for comparison between mobile collected, and in-field static
image capture and residue quantification (Appendix A). Driving speed was maintained between 40-45
km/h to ensure image capture of every field with shutter actuation on each camera set at 5-second intervals.
Following a previous field survey in Elgin County on 2016-05-11, it was determined that a reduced driving
speed be implemented to reduce vibration through the vehicle to the elevated camera platforms. Therefore,
a speed restriction of 40-45 km/hr. was used on the subsequent (Essex and Elgin County – 2016.05.18,
2016.05.20, respectively) acquisitions.
Driving speed and shutter actuation are inherently related, and based on average roadside field dimensions.
The following equation was used to calibrate both shutter actuation and driving speed.

20                    SA = DS (m/sec) / (MFW (m)/3)                    (Equation 1)


Where SA = shutter actuation interval in seconds; DS = vehicle driving speed in metres per second; MFW
= mean field width in metres, based on pre-site planning of field polygon network for each respective
sample area. This value is then divided by 3 to ensure a minimum of 2 usable images for each respective
field. Multiple field image captures are required in event that undesired features (people, vehicles, trees,
etc.) are visible at the forefront of the imaging plane.



**Data Processing**

**Data Transfer and Sorting**

Following each site acquisition, image files were transferred to a desktop computer for sorting. A total of
18,462 images were acquired for the 2 routes. The images were manually sorted by site, date, and look
direction for the particular camera whereby any images not meeting the requirements of the project were
subsequently deleted. Examples of deleted scenes include images of forests, houses, quarries, intersections,
overpasses, non-agricultural structures, road-vehicles, or any that contained identifiable footage of
individuals.

**Geometric Rectification**

An issue prevalent with wide-angle oblique image capture is the so-called fish-eye, or barrel effect. The
geometric distortion in the radial direction, while present in all aspects away from the centroid of the
imaging plane, results from compression in peripheral regions allowing for a wide-angle view to be
presented in the image plane (Kim and Pail, 2015). Such distortion presents issues in many land-use studies
by virtue of a variable pixel-to-ground scale across the image plane, and the removal of lens distortion is
often preferred in multi-class image classification, especially where volumetric measures of scene features
are required (Chow, *et* al., 2018; Stockdale, *et al*., 2015). While not absolutely pertinent to this land-use
study, image correction was performed using Liquivid© Video Fisheye Removal software; other open-
source image correction software, such as Mathmap:GIMP; GML Undistorter; RadCor; Photivio;
FisheyeGL, among others (listoffreeware.com, 2019) are also readily available for the correction of wide
angle lens distortion. The correction permits cross platform image sharing for a subsequent study where
geometrically rectified imagery is required.

**Coordinate Encoding and Verification**

The GPS vehicle mounted cameras include horizontal geographic lat/long coordinate information in
addition to other variables including, elevation, slope and aspect that are transferrable to the imagery via
still-to-video image conversion. Transfer of ancillary information to coordinate geometry was performed by
creating a stop-motion video comprised of the 4 sorted, and geometrically rectified image data-sets (left
and right for 2 acquisitions), then deconstructing the video back to individual static images. The process
allowed for extraction of coordinate coding for each image and the generation of a point for each image
location to be displayed in a GIS environment. In this instance we used ESRI ArcGIS, however the same




methodology may equally be employed using many open source alternatives (e.g. GRASS, WhiteBox,
QGIS, uDIG, etc).

**GIS Integration and Map Overlay**

**Geodatabase creation**

A geodatabase of the imagery was created within an ESRI ArcGIS 10.1 environment. The roadside images
were linked to a point feature class layer for spatial identification based on the inherent coordinate data
captured coincident with the roadside imagery (Figure 3). Determining the point feature classes were then
linked to individual field polygons to allow for photo identification of each field within the study region.
This procedure allows the user to zoom in to each field location and visually assess any roadside images
connected to the given field polygon (Figure 4).

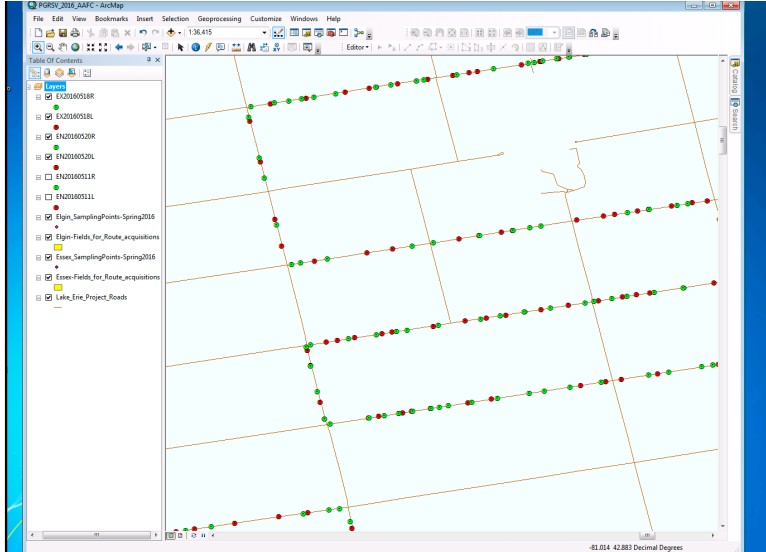

**Figure 3. Geo-referenced roadside survey image locations. Green = right viewing, Red = left viewing to vehicle forward movement. Direction of movement determined by imbedded time-stamp of subsequent left or right geo-referenced points.**



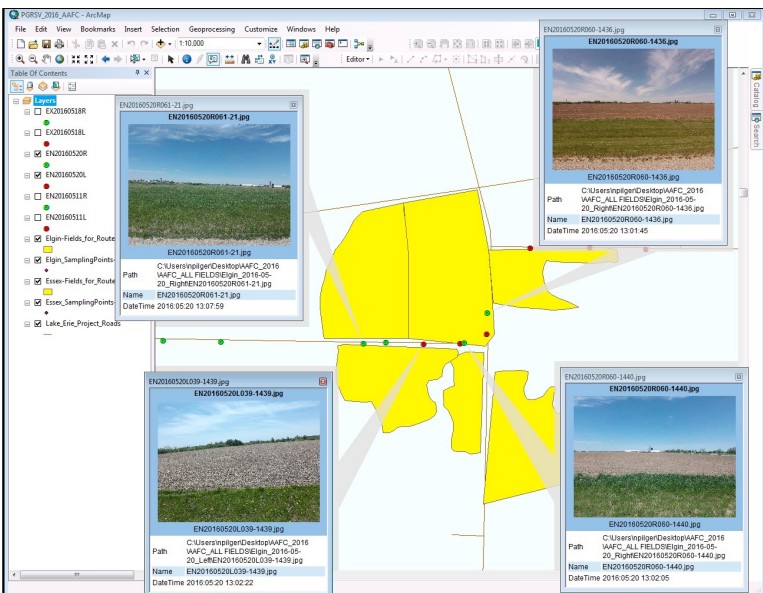

**Figure 4. Hyperlinked image files to point and field polygon network.**

A benefit of this procedure is the ease of updating visual examples of each respective field in a given area, and the immediate comparison to changes in both crop type and tillage practices. The technique is adaptable to be carried out over a series of years, while minimizing the time and costs to be physically present in the reference, or validation fields themselves. Images captured by field research were assessed by OMAFRA staff trained in conducting visual surveys to belong to one of 5 to 6 residue classes, which were then rescaled into the 4 cover classes (Figure 5) to be used in validating the oblique mobile image capture.

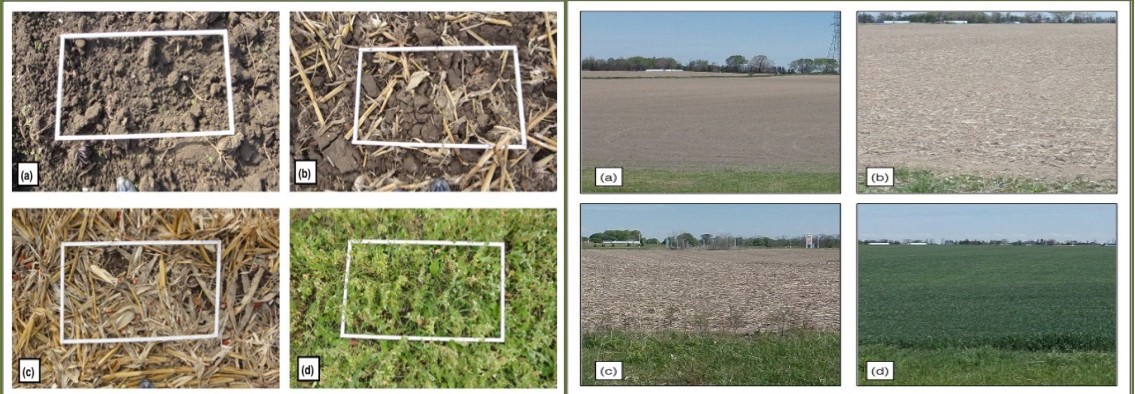

**Figure 5. Classes as recorded using in-field, and roadside survey. (a) Conventional; (b) Conservation; (c) No-Till; and (d) Green Cover.**





Ground-based validation of roadside image capture was performed using data from 114 research sites
across the two counties collected by AAFC scientists coincident with roadside image capture. The in-field
research sites were evaluated for residue percentage using a photo-grid sampling technique where average
counts were derived from random selections on three  digital images captured at 90 degrees from surface
normal, or nadir view over 0.75 x 1.0 meter survey quadrats. Residue and green cover counts of the photos
were performed using of 10 x 10  digital grids, representing 100 points for each imaging frame (Laamrani,
*et al.*, 2017, 2018),  where cover percentage was based on presence or absence over each of the 100 grid
intersection points and categorized as conventional (0% - 30%); Conservation (30% - 60%); and No-Till
(60% - 100%) residue (figure 6). The green cover class was assigned when >90% field was visually
composed of green, actively photosynthesizing vegetation.

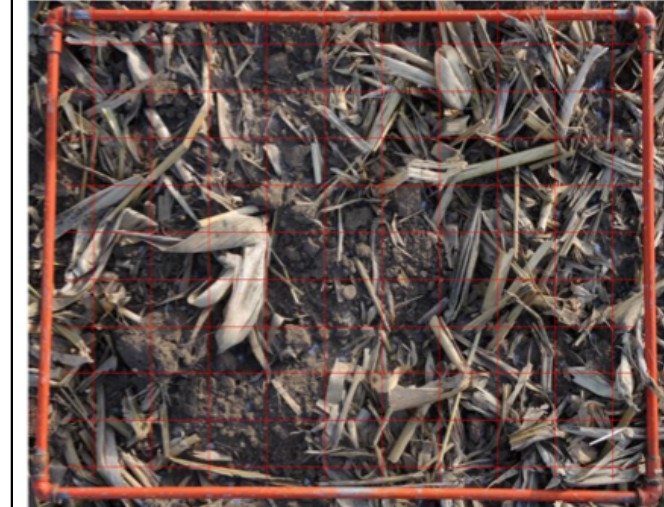

**Figure 6. Superimposed 10 x10 photo grid over corn residue field sample quadrant.**

This digital method of in-field residue evaluation was found to be highly correlated to traditional line
transect methods to accurately estimate post-harvest residue percentages, but in a more efficient manner
(Laamrani *et al.*, 2017).
**Results**
A confusion matrix (Table 2) was used to compare the mobile roadside imagery classes against in-field
collected data classes (Figure 5) for the 114 research sites distributed over the two counties surveyed during
this research project. While overall agreement between roadside collected imagery (RS) and in-field (IF)





measures was strong, there are some minor areas of confusion, primarily between conservation (CS) and
no-till (NT) fields. This may be explained by the visual similarity at distance from the roadside between
these two land-classes (Figure 5 b and c (right side)).
**Table 2. Error matrix mobile roadside (RS) oblique vs ground collected nadir in-field (IF) imagery, for**
**conventional tillage (CV); conservation tillage (CS); no-till (NT); and green cover (GC)  practices. Validation**
**carried out against 114 AAFC field sites in Elgin and Essex County. Omission and Commission error values**
**for each land class are reported in the last row and column respectively, with an overall accuracy of 93%.**

| n = 114 | IF-CV | IF-CS | IF-NT | IF-GC | Total | Error |
|---------|-------|-------|-------|-------|-------|-------|
| **RS-CV** | 30 | 1 | 0 | 0 | 31 | 3.2% |
| **RS-CS** | 0 | 21 | 5 | 0 | 26 | 19.2% |
| **RS-NT** | 0 | 2 | 37 | 0 | 39 | 5.1% |
| **RS-GC** | 0 | 0 | 0 | 18 | 18 | 0% |
| **Total** | 30 | 24 | 42 | 18 | | |
| **Error** | 0% | 12.5% | 11.9% | 0% | | **OA= 93%** |

Another issue relating to confusion between the tillage classes involves the look direction variation
between nadir and oblique image capture. For example, a harvested field of corn will appear to contain
higher levels of residue in an oblique image, as we are viewing the residue from the side, and overlap
occurs as a function of perspective, whereas in a nadir view a greater proportion of bare soil will be visible
within the image. As both Conservation and No-Till meet thresholds representative of conservation
methods (i.e. >30% cover for soil erosion protection), the  misclassification between the two is deemed
acceptable in this case.

**Challenges and Environment**
While few, there are a number of challenges that must be addressed in carrying out an operation of this
scope. These will be addressed in order of technological importance. Weather, is paramount if requiring
coincident optical satellite imagery with field interpretations; besides this requirement, for roadside data
collection the following are the key challenges to overcome.





Driving speed: technically higher speeds should lead to a blurring effect on short distance image capture.
This is especially true when light levels are low. However, in comparative tests following all acquisitions,
shots captures at 40 km/hr in bright clear atmospheric conditions were only marginally sharper than those
captured at 60 km/hr + during overcast conditions.  The real benefit of a reduced driving speed, therefore,
lies in a reduction of vibration from wheel-base through to camera mount.
Wind speed: as with vehicular velocity above, higher wind speeds increase vibration through the vehicle
and the extended camera mounts. Vibration can significantly affect image clarity and subsequent
assessments. From this pilot project there is evidence in image capture footage where heavy gusts can also
offset the horizontal plane of the camera. While efforts were in place to minimize such effects (insulated,
damping core wire ties from cameras and extension poles to vehicle roof-rails) excessive gusting can affect
the horizontal image plane.
Shutter actuation: Shutter actuation, is intrinsically linked to driving speed, and external light conditions.
While one could capture images at up to 10 shots/sec. the associated image sorting time required if
conducted manually as in this project would render such operation financially unsound. Setting camera
aperture to ensure that every field is captured a minimum of 3 times in a pass required pre-assessment of
field orientation, and dimensions as this will vary by the land survey system by geography. This enabled
calculation of maximum driving speed  (in this case it varied between 30-50 km/hr) with a 3-5 second
actuation interval.

Privacy concerns: Though listed last, privacy concerns are paramount in post production though not a
technical issue. Any images where people, especially children; vehicles; homes; etc. are visible and
identifiable have potential to raise privacy concerns and cannot legally be distributed through any
commercial or government shared channels. With a sound editing and sorting methodology, any such
images would be deleted in an expedient manner, following a best practice protocol.

**Conclusions**
Key benefits of the mobile ground-based reference data collection method described in this paper are the
flexibility garnered through not requiring any specific meteorological conditions, as is the case with most
optical airborne, and orbital platforms; enhanced safety by removing manual real-time roadside
classification, and driver distraction by incorporating an automatic imaging system.; and creating archival



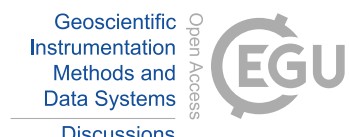

footage on land and lands conditions for future reference. Temporally, collection dates are limited in tillage
assessment studies carried out in temperate climates. Ground-truth imagery is required with in the late
autumn, prior to snowfall; or in late spring, following snowmelt, but prior to any seedbed preparation
practices carried out prior to planting. In southwestern Ontario, skies are generally overcast for the majority
of days in these narrow temporal windows, thus limiting traditional classification using orbital, or high
altitude airborne image capture. Specific fields that have not been harvested at the time of roadside image
capture, may readily be re-visited, or such field-data may be in-filled at a later date, either via direct contact
with land holders, or through other citizen provided reference data (Foody, 2015). Mounted with opposing
side viewing cameras, the roadside survey vehicle was shown to be highly efficient in the collection of geo-
referenced imagery of up to 500 fields per hour, with an overall level of agreement to in-field ground
surveys at 93 percent, employing a single vehicle, and 1-2 operators, producing a more reliable and robust
data-set for extrapolation to larger areas, compared to the 5 to 10 fields which could be surveyed in the
same time frame using conventional in-field methods.
Another benefit is that time series of these surveys would permit change detection analysis over subsequent
years to evaluate climate adaptations, routine monitoring for productivity, soil surveys and yield estimates
(Fisette, *et al*., 2014, Kennedy, *et al*., 2009); to determine impacts of regulations that may result in the
adoption of particular cropping practices in the province (Vercammen, 2011); and to inform recommended
methodologies that can be used by industry, provincial or federal organizations for more routine
measurement of soil cover.

This project demonstrated that this method can provide rapid determination and dissemination of post-
harvest tillage and green cover practices over county level areas even where atmospheric conditions are
unfavourable for satellite remote sensing, while improving on financial, temporal, and safety costs for in-
field verification data acquisition.

While this study focused on the resource-based utility of employing mobile roadside image capture for use
in characterizing post-harvest field conditions, a follow-up project is underway to investigate residue
decomposition rates and subsequent field operations to determine optimal timing in such mobile field
surveys. Also, based on date of image capture, some fields showed evidence of recent soil turning,
therefore the fields analyzed may indicate higher levels of soil disturbance being classed as conventional
tillage than what may in fact be a conservation practice carried out by the landowner. In this respect, one
must take into account the temporal influence of any tillage residue survey. With many agricultural surveys



being carried out coincident with satellite overpass, potential classification errors, as described in this manuscript, stress the benefit of a non-orbital dependent classification technique.

**Acknowledgements**

This research was supported by funding from the Ontario Ministry of Agriculture, Food and Rural Affairs and Agriculture and Agrifood Canada. Thanks to Ahmed Laamrani, for compiling driving route maps; AAFC field scientists for ground verification data; and to Profile Geomatics, Canada, for providing the use of their roadside survey vehicle and imaging equipment. Additional software and hardware support was provided by the Department of Geography, University of Guelph, Canada.

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









3    **Appendix A.**

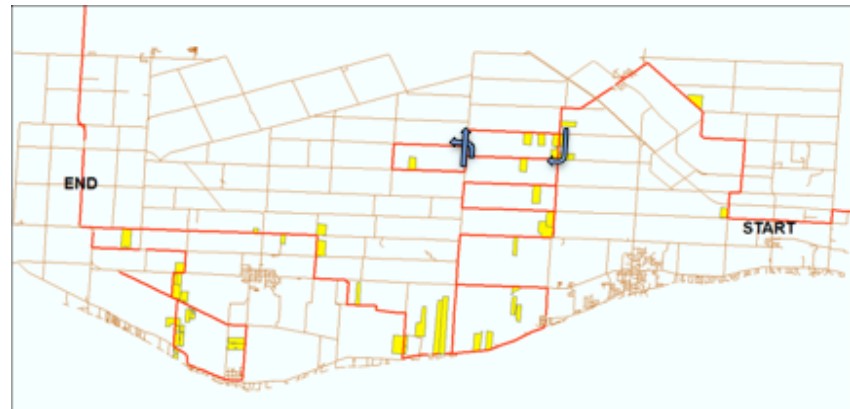

6    Figure A1. Essex County driving route map. AAFC fields denoted in yellow.

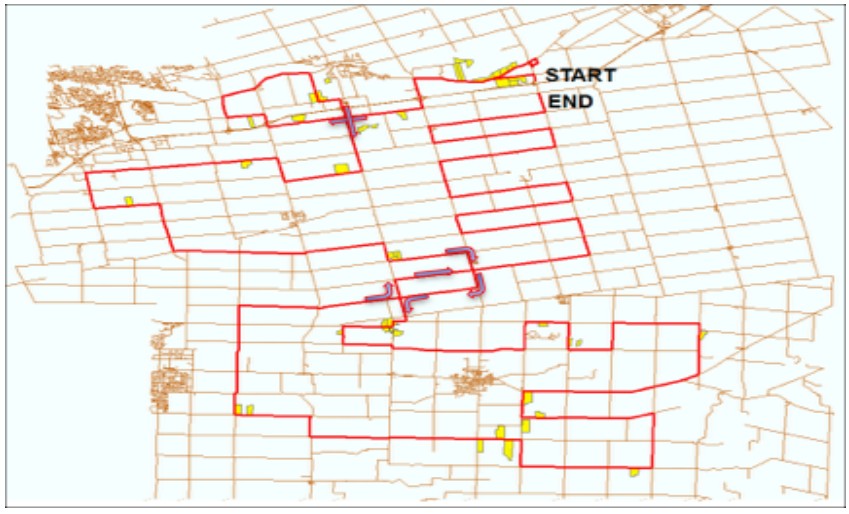

10    Figure A2. Elgin County driving route map. AAFC fields denoted in yellow.