# Peer review of "Semi-automated roadside image data collection"

_Geoscientific Instrumentation, Methods and Data Systems, 2019_

## Referee Comment (RC1) · Anonymous Referee #1 · 26 Aug 2019

This article reports on a system to acquire ground reference data on crop fields from a car mounted side-looking set of cameras. The work is interesting and seems to focus on an operational application of governmental/environmental agencies. The paper is clear, easy to read and focused on a real-world problem. There are, however, a set of concerns:

A fuller discussion on training and testing data acquisition is needed in the introduction. In some instances a small number of geo-referenced photos may be adequate but this is far from a complete discussion. Issues of sampling (notably the sample size and design) and basic data collection (the spatial grain) etc. need fuller discussion. Sample design is a key issue that needs fuller discussion in the article as the approach used is not a probability sample. As such it may meet many needs but not all (e.g. as

noted below it would be an unsuitable sample to use in image classification accuracy assessment).

The description of the test site must make it clear that CS sites are also present - this class is not mentioned yet its presence is critical to the study.

In the data processing the discussion is very vague. It would be really useful, for example, to know how many photographs were retained and used.

The data processing section needs to address aspects of privacy (which do get mentioned late on the paper on p12). The system really needs to be able to acquire data in a way such that faces and items such as car licence plates are blurred out. This type of processing is quite basic in systems that seek to ensure GDPR compliance.

More details on the labelling of photographs is required. Were photographs labelled by just a single person? In many studies labelling is often based on 3+ annotators – allowing basic consensus approaches to be used as well as flagging uncertain cases.

The authors deem their results to be acceptable. They may well be but it should be based on strong evidence and reasoning. What level of error is tolerable? The authors also need to recognize that the accuracy of the system will vary with relative abundance of the classes, accuracies in the order 80-100% seem quite feasible. Is the lower value stlll acceptable?

The work in many ways is an automated version of a basic windshield survey that has been used for decades. Why not compare to a windshield survey – a qualified individual sitting as a passenger in a car would probably be very accurate.

There are a variety of obvious issues that are not addressed well. These range from clear concerns such as the presence of hedges/fences that obscure view to problems of inter-cropping. These probably do not crop up much in the test site but deserve mention.

A potential problem with the work is the reader making an innocent mistake and thinking the system would be useful as a source of testing data for validating/accuracy assessment of satellite image classifications. This system is not (and the authors do not claim it to be). It should still be recognised that good practices for validation call for a probabilistic sample – the sample acquired by the system does not meet this – it is biased (to roadside locations) and unrepresentative. This issue should perhaps be noted simply to stop an interested reader making a mistake.

I cannot help wonder – why use this system and not a basic, nadir viewing, photographs from a drone/UAV? Cheap, easy to use from roads etc. and is nadir view that fits with the standard ground data. Surely a UAV based camera system offers a better outcome?

Minor issues: - The term 'corroboration data' is unclear – perhaps use 'ground reference data'? - Clarify the last sentence of the introduction – what is the role of the orbital data? - It is awkward to refer to 'RS' in Table 2. On superficial reading this might be interpreted as meaning remotely sensed (e.g. from satellite). Need to be clear this is from the car-based camera system.

---

## Referee Comment (RC2) · Anonymous Referee #2 · 15 Jan 2020

This paper describes the development of a mobile agricultural practice inspection system using images taken from the side of a motor vehicle. Such a system seems to be of some applicative interest for agricultural administration and statistics. However, there are many on-board shooting vehicles and the development of a new one, in itself, is not a very original contribution. Moreover, the paper, although pleasantly styled and easy to read, is actually quite poorly written and ultimately unclear.

For example, reading the abstract and the introduction, one might think that the method produces data useful for the calibration or verification of remote sensing methods, but this is not the case. In reality, the proposed system is more of an alternative to remote sensing and high-altitude aerial imagery. One might have expected to see a comparison between the results of the system and those of the alternative methods, but this is

not the case.

Later in the paper, it is understood that the system is supposed to replace a traditional surveying method, by operators on board of vehicles. One might therefore expect to see a comparison, in terms of efficiency (acquisition & processing time) and quality of the result, but again, this is not the case.

The only experiment shown compares interpretations (apparently purely visual) made on images taken in the fields (in vertical view) with images in oblique view obtained with the help of the proposed mobile system. Finally, the effect of the angle of view in this experiment is largely evaluated, rather than the system itself.

The experimental protocol is not well described. Someone unfamiliar with the classification methods used by the operators (especially for oblique views) has difficulty understanding them and there is no bibliographic reference on this point. One of the practical interests of the system would be to automate the analysis task, but this is not discussed at all, which is a pity.

The only equation in the paper is wrong or, at the very least, poorly commented on. A speed being the ratio of a distance to a time, SA, defined as the ratio of a speed to a distance, cannot be a time, but rather a frequency. In addition, it is disturbing to talk about shutter actuation because one would tend to associate this term with the speed at which the shutter closes. This one, with the speed of the vehicle and the aperture of the diaphragm, condition the motion blur present in the images: vibrations are not the only possible sources of image quality loss. By the way, what is the type of camera shutter (global or rolling shutter)? It is not clear how the field of view is estimated. Are the cameras geometrically calibrated?

It would still be interesting to argue a little more about the justification for the choice of carrier. Rather than a land vehicle, with the drawbacks that this entails (oblique viewing, vibrations, etc.), one could have considered imagery by drone, Ultra-Light plane or helicopter. This also makes it possible to pass under the clouds, and that

would especially offer very interesting efficiency, while remaining in vertical sight.

From a paper structuring point of view, it is annoying that the purpose of the work only appears at the bottom of page 3. It would be better to state them at the beginning of the introduction, before giving details of the application, making a critical inventory of the existing means of inspection and positioning the choices made. In addition, it is usual to end the introduction by describing the structuring of the paper (announcement of the plan).

For all these reasons, I cannot support the publication of this paper, at least in its present state.

---

## Author Comment (AC1) · 12 Feb 2020

RC: This article reports on a system to acquire ground reference data on crop fields from a car mounted side-looking set of cameras. The work is interesting and seems to focus on an operational application of governmental/environmental agencies. The paper is clear, easy to read and focused on a real-world problem. There are, however, a set of concerns:

RC: A fuller discussion on training and testing data acquisition is needed in the introduction. In some instances a small number of geo-referenced photos may be adequate but this is far from a complete discussion. Issues of sampling (notably the sample size

and design) and basic data collection (the spatial grain) etc. need fuller discussion.

AC: The goal of this study was to provide a census, not a sample. Every field was imaged, so the sample size was 100% (less those fields that were obscured by trees or hedges - and those numbers are few -less than 1% of the area fields). This point will be clarified in the introduction and again in the methods section.

RC: Sample design is a key issue that needs fuller discussion in the article as the approach used is not a probability sample. As such it may meet many needs but not all (e.g. as noted below it would be an unsuitable sample to use in image classification accuracy assessment).

AC: This study was an area wide (county level) census, not a sample. In follow up studies we are separating the fields imaged randomly and using 50% for calibration and 50% for validation. This study was focused as a technical letter for a description of an efficient and cost effective process for collecting ground reference data (1 person vs 30; 5000 fields vs 50, one vehicle vs six, 5 hours vs 12, etc.). As with the previous comment, this will be further clarified in both the introduction, and in the discussion portion of the paper.

RC: The description of the test site must make it clear that CS sites are also present - this class is not mentioned yet its presence is critical to the study.

AC: Conservation (CS) tillage sites are mentioned in the paper, combined with No-tillage sites, they make up roughly 60% of the harvested fields surveyed. I do see the confusion here, to remedy this we will include a Totals table with both number of fields in each class as well as overall percentages. Thanks.

RC: In the data processing the discussion is very vague. It would be really useful, for example, to know how many photographs were retained and used.

AC: These values were included along with the new table as described in response to your previous comment.

RC: The data processing section needs to address aspects of privacy (which do get mentioned late on the paper on p12). The system really needs to be able to acquire data in a way such that faces and items such as car licence plates are blurred out. This type of processing is quite basic in systems that seek to ensure GDPR compliance.

AC: I agree, that from a commercial standpoint this would be useful - the blurring in Google Car imagery, however, is performed post data collection using recognition (AI) software. As this was a pilot research project (not designed for commercial dissemination), instances where houses, people or vehicles were imaged were simply discarded. The rate of image capture was such that 2-3 images would be captured of every property. We were not interested in house, only post harvest agricultural fields. Therefore, the only images that were of other vehicles or people, were captured at intersections, or when passing through residential and/or commercial areas on the outskirts of the agricultural area - these images were of no use to the study, so were deleted during primary sorting following data capture.

RC: More details on the labelling of photographs is required. Were photographs labelled by just a single person? In many studies labelling is often based on 3+ annotators – allowing basic consensus approaches to be used as well as flagging uncertain cases.

AC: Post sorting and deletion of non-agricultural field imagery, all subsequent images were visually classified by 4 individuals and recorded on spreadsheets. Contrary classifications were allocated their final classification based on simply majority. For the 4 classes, confusion was primarily between Conservation (CS), and No-Till (NT) classes where quantification of residue was close to 60% (the division between the two classes). Cover crops were green, and Conventional tillage was lacking any significant residue, therefore mis-classification of these categories was virtually non-existent. The confusion between Conservation and No-Till, was also slight, and considered acceptable, as both are actually conservation tillage methods. Further clarification on this matter should have been included, and was addressed in the subsequent draft.

RC: The authors deem their results to be acceptable. They may well be but it should be based on strong evidence and reasoning. What level of error is tolerable? The authors also need to recognize that the accuracy of the system will vary with relative abundance of the classes, accuracies in the order 80-100% seem quite feasible. Is the lower value stlll acceptable?

AC: As addressed in response to the previous comment, error was deemed acceptable as it dealt with confusion between upper levels of tillage in Conservation classes (30-60% residue) and No-Tillage (60% + residue) classes. Both represent conservation tillage, with very similar end results. This was clarified again in the conclusions.

RC: The work in many ways is an automated version of a basic windshield survey that has been used for decades. Why not compare to a windshield survey – a qualified individual sitting as a passenger in a car would probably be very accurate.

AC: The rationale behind not employing a standard windshield survey is in human error, missed fields, reduction is vehicle speed to permit GPS and class recording, and inability to simultaneously record information from both sides of the roadway. Another key benefit is the production of hard data which may be re-examined at a later date, or employed in change detection studies. Finally, having one or (better) two passengers recording information requires additional personnel, which increases data collection costs. This study was based on securing the greatest amount of data in the most efficient and cost effective manner, hence a single driver, and multiple cameras automatically capturing imagery at 3-5 second intervals.

RC: There are a variety of obvious issues that are not addressed well. These range from clear concerns such as the presence of hedges/fences that obscure view to problems of inter-cropping. These probably do not crop up much in the test site but deserve mention.

AC: The issue of intercropping was not an issue for the fields imaged by the roadside survey vehicle, however, we agree that some mention should be made for this case. For

the counties chosen for post-tillage residue assessment, fields are planted in annual rotations (for the most part), consisting of corn, soybean, or winter wheat. As for concerns relating to features obscuring camera view, these were images were discarded dduring initial sorting/classification along with non-agricultural field images, although the roadside vehicle does carry a pair of UAV's, so realistically those obscured fields could be imaged, if it were not for concerns discussed in response to your comment regarding UAV systems below. The cameras were automated to capture an image at set temporal increments (e.g. every 5 seconds), so along with duplication from when the vehicle was at a stop, such images were extracted from further classification. We suspect overall we conducted a 99% census of post harvest tillage residue in the region of study. Further clarification on the number of images removed during initial sorting will be included in the subsequent draft.

RC: A potential problem with the work is the reader making an innocent mistake and thinking the system would be useful as a source of testing data for validating/accuracy assessment of satellite image classifications. This system is not (and the authors do not claim it to be). It should still be recognised that good practices for validation call for a probabilistic sample – the sample acquired by the system does not meet this – it is biased (to roadside locations) and unrepresentative. This issue should perhaps be noted simply to stop an interested reader making a mistake.

AC: We have included a comment to this effect. However we also argue that this study serves as a potential surrogate to traditional ground reference data collection which is costly, inefficient, potentially dangerous, and involves property trespass. Relative to our ground survey data our the classification of the vehicle mounted images provides acceptable classification accuracy. Further the data collected by this roadside survey method captures every field in the study region (given the structure of fields and roads in this region).

RC: I cannot help wonder – why use this system and not a basic, nadir viewing, photographs from a drone/UAV? Cheap, easy to use from roads etc. and is nadir view that

fits with the standard ground data. Surely a UAV based camera system offers a better outcome?

AC: Yes, a UAV would be more versatile with the ability capture both oblique and Nadir image data, in RGB or False Colour format. The vehicle used in this project actually carries two UAVs in the back. The main issue with running them as opposed to the roadside survey as performed lies in the limited range, and time requirements to operate drones within the legal requirements put forth by Transport Canada. Excepting extreme circumstances (e.g. search and rescue operations) Drones in Canada must be down line-of-sight (LOS) only. Additionally there are restrictions barring operation without Special Flight Operations Certification (SFOC) within 100 metres of any person, vehicle, occupied structure, and livestock. Barring such legal restrictions, a stop-and-go survey of every field at a county-level scale would be no more efficient than physical trespass into the fields as is the current method for generating a sample of tillage practices. This study was designed not to sample, but to provide a large area census with minimal costs, and personal. The reason why the vehicle does carry UAVs, was to address one of your previous concerns regarding the obstruction of clear lines of sight from hedges/fences which, while not numerous are still prevalent in the study areas we were looking at. However, to facilitate maximum coverage in minimal time, such obscured field images were removed during the preliminary sorting/classification phase. Mention of this issue was implemented in the subsequent draft.

RC: Minor issues: - The term 'corroboration data' is unclear – perhaps use 'ground reference data'? -

AC: Agreed. This has been updated as suggested

RC: Clarify the last sentence of the introduction – what is the role of the orbital data? -

AC: Removed 'orbital data' which referred to unsupervised classification of Landsat imagery; and replaced with: "The survey system was developed to compare to in-field photos for measuring soil cover in order to determine the value of roadside acquisition

for both routine ground truth data collection for remote sensing analysis of soil cover, and the utility of using such data collection as a surrogate to standard practices which are both costly and inefficient to implement".

RC: It is awkward to refer to 'RS' in Table 2. On superficial reading this might be interpreted as meaning remotely sensed (e.g. from satellite). Need to be clear this is from the car-based camera system.

AC: We agree, this acronym was altered to avoid mis-interpretation.

---

## Author Comment (AC2) · 12 Feb 2020

RC: This paper describes the development of a mobile agricultural practice inspection system using images taken from the side of a motor vehicle. Such a system seems to be of some applicative interest for agricultural administration and statistics. However, there are many on-board shooting vehicles and the development of a new one, in itself, is not a very original contribution.

AC: I agree there are a host of mobile mapping vehicles commercially available for hire - many with 3D mapping (LiDAR). This project, however, was based around minimizing costs (labour, time, resources) for the capture of ground verification data of post-

harvest tillage practices only over large geographical regions. Key issues included the timeliness of acquisitions to match satellite coverage and the period of time when land management assessments are necessary. It was envisioned as a surrogate, or direct replacement for traditional methods which employ upwards of 30 people, require six or more vehicles, and gather information on tillage residue, and tillage operations over rather broad classes. The results of which (40 - 60 fields) are then extrapolated across a county-wide region to determine the percentage of agriculture operations which are shifting from conventional (full soil overturning and grooming post harvest) to conservation (residue left behind) and/or cover crop operations (note that No-till operations are another form of conservation tillage). Such traditional methods employed in securing such ground reference data may also involve access onto priviate property which requires significant resources prior to the campaign to gain the necessary permissions, potential safety risks (crossing culverts and so forth), and limited samples which may prove not to be actually representative of the larger region. This project - a precursor to follow-up studies which will be using our data in validating satellite classifications, was not designed to sample, but to provide a cost effective census of field conditions on a particular day following the majority of harvest operations.

RC: Moreover, the paper, although pleasantly styled and easy to read, is actually quite poorly written and ultimately unclear. For example, reading the abstract and the introduction, one might think that the method produces data useful for the calibration or verification of remote sensing methods, but this is not the case. In reality, the proposed system is more of an alternative to remote sensing and high-altitude aerial imagery.

AC: The paper has been re-written sections to improve clarity. In certain cases we do see this as an opportunity to either form a calibration/validation data set for remote sensing imagery, but as noted during the small time windows when this data is necessary, techniques that are accurate enough to provide a census and therefore a replacement of optical remote sensing data may be necessary. Notably, a random sample of the census data from this project is being utilized as ground verification training

data (calibration data) for classification of both optical and radar orbital imagery with the remaining data being used for post-classification verification. The argument presented in this paper is that the census data may be employed for classification on its own merits - for a rather broad 4-5 category classification. Both of these are follow-up studies, however, which explains why there was not significant discussion on the utility of this data collection method. The subsequent draft of this manuscript will address such issues in a clearer manner.

RC: One might have expected to see a comparison between the results of the system and those of the alternative methods, but this is not the case. Later in the paper, it is understood that the system is supposed to replace a traditional surveying method, by operators on board of vehicles. One might therefore expect to see a comparison, in terms of efficiency (acquisition & processing time) and quality of the result, but again, this is not the case.

AC: I feel that perhaps a table specifying the comparison between the roadside survey method and those traditionally employed would serve to clear this up. The issues you refer to are present in the paper, but should be more concisely arranged. For example 5000 fields versus 50; one operator versus 30; one vehicle versus six; 5-6 hours of data collection versus 12; etc. Also imagery not used (duplicates, non-agricultural imagery, etc.) should also be included. The benefits is gaining census rather than sample data, yet that is also the detriment, as there is a lot of data produced.

RC: The only experiment shown compares interpretations (apparently purely visual) made on images taken in the fields (in vertical view) with images in oblique view obtained with the help of the proposed mobile system. Finally, the effect of the angle of view in this experiment is largely evaluated, rather than the system itself.

AC: More detail is provided in this respect, and less on the issues related to oblique versus nadir view. Post sorting and deletion of non-agricultural field imagery, all subsequent images were visually classified by 4 individuals and recorded on spreadsheets.

Contrary classifications were allocated their final classification based on simple majority. For the 4 classes, confusion was primarily between Conservation (CS), and No-Till (NT) classes where quantification of residue was close to 60% (the division between the two classes. Cover crop was green, and Conventional tillage was lacking any significant residue, therefore mis-classification of these categories was virtually non-existent. The confusion between Conservation and No-Till, was also slight, and considered acceptable, as both are actually conservation tillage methods. Further clarification on this matter are addressed in the subsequent draft.

RC: The experimental protocol is not well described. Someone unfamiliar with the classification methods used by the operators (especially for oblique views) has difficulty understanding them and there is no bibliographic reference on this point.

AC: The classification methodology was expanded upon as described above.

RC: One of the practical interests of the system would be to automate the analysis task, but this is not discussed at all, which is a pity.

AC: A follow-up study performs exactly what is proposed. We are using Semantic segmentation and Deep Learning (DL) to automatically classify such roadside imagery. Much in the same fashion as autonomous vehicles classify in near-real-time objects and entities around them, this DL segmentation protocol classifies fields based on tone and texture into 4 classes (conventional tillage, conservation tillage, no-tillage, and cover crop) all other scene entities are discounted as 'other' and effectively removed (blacked out) from the image scene. We hope in time to have a system that will auto-classify, so at the end of a survey a final classified product is ready for dissemination. This paper, however was meant to describe how the camera system is developed as cost effective efficient data collection technique, which captures 100% of the fields in a given county area. Perhaps a note to this effect could be included at the end of the manuscript in discussion to follow-up studies.

RC: The only equation in the paper is wrong or, at the very least, poorly commented

on. A speed being the ratio of a distance to a time, SA, defined as the ratio of a speed to a distance, cannot be a time, but rather a frequency. In addition, it is disturbing to talk about shutter actuation because one would tend to associate this term with the speed at which the shutter closes. This one, with the speed of the vehicle and the aperture of the diaphragm, condition the motion blur present in the images: vibrations are not the only possible sources of image quality loss.

AC: The equation listed relates to programming the camera shutter interval to ensure a minimum of 2 usable images captured for every field in the county study area. The roadside survey vehicle described in this paper was used over three different counties (two mentioned in this paper). To ensure that every single field is adequately imaged, the roadside width (average frontage) of the fields in the county area is required, and an upper limit driving speed. This equation then indicates the time (in seconds) it takes to drive between the two outer boundaries of the average field in the area. With a desire to capture 2-3 images for each field, the mean field width is divided by 3. This does not mean that the vehicle will maintain a steady velocity, as there are always exceptions where a field may be split, or other features (e.g. streams, hedgerows, etc.) may result in narrower plated plots. It is, however a guideline for setting the camera intervalometers, as they are external to the vehicle and not readily accessible while in motion. In event of narrow fields, or fields slightly obscured by trees, the operator can slow down, stop, and/or double back to ensure adequate coverage. As to your second point about the camera itself, such details can be included in the subsequent draft of the manuscript. The cameras used are Garmin VIRB XE units that shoot 12 MP images - exposure time is not static, as it is digitally based on incident light saturation, however, in most instances it is operating between 1/1000 and 1/2500 of a second. The cameras are mirrorless and do not have physical shutters or diaphragms but are electronically stabilized to minimize blur. Regardless the equation was omitted from the final draft.

RC: By the way, what is the type of camera shutter (global or rolling shutter)? It is not

clear how the field of view is estimated. Are the cameras geometrically calibrated?

AC: Camera specifications have been addressed in response to the previous comment. The cameras have internal GPS which is auto-calibrated prior to each field operation, and again following battery changes. location error is consistent with any other hand-held Garmin GPS unit operating at 10Hz - generally accurate to within 20cm at variable speeds. More than accurate enough for agricultural field identification.

RC: It would still be interesting to argue a little more about the justification for the choice of carrier. Rather than a land vehicle, with the drawbacks that this entails (oblique viewing, vibrations, etc.), one could have considered imagery by drone, Ultra-Light plane or helicopter. This also makes it possible to pass under the clouds, and that would especially offer very interesting efficiency, while remaining in vertical sight.

AC: While an ultralight aircraft or helicopter would be prohibitively expensive for this type of study, a UAV would be more versatile with the ability capture both oblique and Nadir image data, in RGB or False Colour format. The vehicle used in this project actually carries two UAVs in the back. The main issue with running them as opposed to the roadside survey as performed lies in the limited range, and time requirements to operate drones within the legal requirements put forth by Transport Canada. Excepting extreme circumstances (e.g. search and rescue operations) Drones in Canada must be down line-of-sight (LOS) only. Additionally there are restrictions barring operation without Special Flight Operations Certification (SFOC) within 100 metres of any person, vehicle, occupied structure, and livestock. Barring such legal restrictions, a stop-and-go survey of every field at a county-level scale would be no more efficient than physical trespass into the fields as is the current method for generating a sample of tillage practices. This study was designed not to sample, but to provide a large area census with minimal costs, and personal. The reason why the vehicle does carry UAVs, was to address the obstruction of clear lines of sight from hedges/fences which, while not numerous are still prevalent in the study areas we were looking at. However, to facilitate maximum coverage in minimal time, such obscured field images were removed during

the preliminary sorting/classification phase

RC: From a paper structuring point of view, it is annoying that the purpose of the work only appears at the bottom of page 3. It would be better to state them at the beginning of the introduction, before giving details of the application, making a critical inventory of the existing means of inspection and positioning the choices made.

AC: The purpose of the manuscript was moved as suggested.

RC: In addition, it is usual to end the introduction by describing the structuring of the paper (announcement of the plan). For all these reasons, I cannot support the publication of this paper, at least in its present state.

AC: Yes, this is traditionally standard, and the response to your previous comment would include such.